# Assessing Combined Effects for Mixtures of Similar and Dissimilar Acting Neuroactive Substances on Zebrafish Embryo Movement

**DOI:** 10.3390/toxics9050104

**Published:** 2021-05-06

**Authors:** Afolarin O. Ogungbemi, Riccardo Massei, Rolf Altenburger, Stefan Scholz, Eberhard Küster

**Affiliations:** 1Department of Bioanalytical Ecotoxicology, UFZ—Helmholtz Centre for Environmental Research, Permoserstraße 15, 04318 Leipzig, Germany; rolf.altenburger@ufz.de (R.A.); stefan.scholz@ufz.de (S.S.); eberhard.kuester@ufz.de (E.K.); 2Institute for Environmental Sciences, University of Koblenz-Landau, Fortstraße 7, 76829 Landau, Germany; 3Department of Effect-Directed Analysis, UFZ—Helmholtz Centre for Environmental Research, Permoserstraße 15, 04318 Leipzig, Germany; riccardo.massei@ufz.de

**Keywords:** mixture toxicity, neurotoxicity, antagonism, organophosphate, acetylcholinesterase inhibitors, GABA, behavior, risk assessment, spontaneous movement activity

## Abstract

Risk assessment of chemicals is usually conducted for individual chemicals whereas mixtures of chemicals occur in the environment. Considering that neuroactive chemicals are a group of contaminants that dominate the environment, it is then imperative to understand the combined effects of mixtures. The commonly used models to predict mixture effects, namely concentration addition (CA) and independent action (IA), are thought to be suitable for mixtures of similarly or dissimilarly acting components, respectively. For mixture toxicity prediction, one important challenge is to clarify whether to group neuroactive substances based on similar mechanisms of action, e.g., same molecular target or rather similar toxicological response, e.g., hyper- or hypoactivity (effect direction). We addressed this by using the spontaneous tail coiling (STC) of zebrafish embryos, which represents the earliest observable motor activity in the developing neural network, as a model to elucidate the link between the mechanism of action and toxicological response. Our objective was to answer the following two questions: (1) Can the mixture models CA or IA be used to predict combined effects for neuroactive chemical mixtures when the components share a similar mode of action (i.e., hyper- or hypoactivity) but show different mechanism of action? (2) Will a mixture of chemicals where the components show opposing effect directions result in an antagonistic combined effect? Results indicate that mixture toxicity of chemicals such as propafenone and abamectin as well as chlorpyrifos and hexaconazole that are known to show different mechanisms of action but similar effect directions were predictable using CA and IA models. This could be interpreted with the convergence of effects on the neural level leading to either a collective activation or inhibition of synapses. We also found antagonistic effects for mixtures containing substances with opposing effect direction. Finally, we discuss how the STC may be used to amend risk assessment.

## 1. Introduction

Chemicals typically occur as mixtures in the environment and hence, organisms are exposed to a combination of these chemicals. However, prospective risk assessment is conducted for single chemicals and may not account for combined effects [1]. Since it is practically impossible to test all the possible combinations of chemical exposure, modeling of mixture toxicity allows one to at least predict an expected effect of several chemicals from their individual effects.

Two common mixture toxicity models are concentration addition (CA) and independent action (IA). CA is based on the notion that mixture toxicity can be predicted by the addition of the fractions of exposure and effect concentrations for the mixture components. In addition, the single components of the mixture should cause a similar effect or target a similar receptor in the organism [2]. On the other hand, IA may be applied when compounds are acting independently [3] which has been interpreted as acting on different target sites in the organism [4]. Both models have been found to be reasonably predictive in several studies exposing unicellular organisms to bioactive compounds with known mechanisms of action [5,6,7]. Nevertheless, these models cannot predict the interaction of chemicals at the physical, toxicokinetic or toxicodynamic level [8]. In this case, CA and IA models may be used to evaluate observations as antagonistic (less effect than predicted) or synergistic (higher effect than predicted) and to quantify such deviations. 

Neuroactive chemicals are often found in insecticidal and pharmaceutical products in which they represent active ingredients designed to interact with specific targets and receptors of the nervous system. Busch et al. [9] found that neuroactive substances are the largest group (13%) of chemicals detected in European surface waters. Despite neuroactive substances being often detected in the environment, only a few studies have explored how neuroactive substances act in mixtures to induce combined neurotoxicity (e.g., Corbel et al. [10]; Yang et al. [11]) and how to use the mode of action knowledge to group them for mixture effect prediction using CA and IA models. 

Zebrafish embryos are considered as an alternative model to animal testing since they are considered to feel less pain or distress [12]. Due to behavioral patterns already established in embryonic stages, embryos are also frequently used as a model for neurotoxicity assessment. Several behavioral test methods have been developed such as spontaneous tail coiling (STC), photomotor response (PMR) and locomotor response (LMR) (reviewed in Ogungbemi et al. [13]). Despite the potential of non-lethal endpoints such as behavior for ecotoxicology research, the applicability of CA and IA models to such endpoints for mixture effect prediction is not well studied. Hence, it is valuable to investigate the applicability of CA and IA models for such experimental systems to predict and understand how mixtures of neuroactive substances may act in the environment. To implement mixture models, bioassays capable of quantitatively detecting impact on the nervous system are required. In this study we explored the spontaneous tail coiling (STC) of zebrafish embryos, one frequently used assay for assessing neuroactivity. STC represents the earliest motor activity observed in developing zebrafish embryos. It is the result of the innervation of the muscles by the primary motor neurons and can be observed beginning at 17 hours post-fertilization (hpf) [14,15]. Measurement of the STC frequency has been proposed as an indicator of adverse effects on the function and development of the nervous system which could lead to population and ecosystem effects [13,16]. Consequently, the STC has been used to study the effects of diverse neuroactive chemicals [17,18,19,20]. Until now the STC has not been used as a test method to measure mixture neurotoxicity based on a chemical’s mode or mechanism of action. In this study, we define the mechanism of action as the interaction of neuroactive chemicals with specific molecular targets such as acetylcholinesterase (AChE) and gamma aminobutyric acid (GABA) activated ion channels. On the other hand, mode of action is defined here as the series of key events (including the mechanism of action) in the nervous system leading to a measurable toxicological response such as hyper- or hypoactivity behavior phenotypes (referred to as effect direction onwards). Hypoactivity refers to a decrease in the STC frequency, while hyperactivity refers to the increase with respect to the level in non-exposed embryos.

The STC test has been shown to discriminate movement activity changes due to exposure to chemicals with different modes of action causing either hyper- or hypoactivity but not those with different mechanisms of action [13,17]. Based on previous results in Ogungbemi et al. [13,17], we postulate the STC neuroactivity hypothesis which states that a neuroactive substance will induce increased STC (hyperactivity) in zebrafish embryos if its mechanism of action directly or indirectly leads to activation of the neuronal synapse and vice versa for hypoactivity. For example, different mechanisms of action such as AChE inhibition and GABA antagonism may both enhance neuronal activation potential in the neuromuscular synapses by inducing the inflow of sodium ions and blocking the inflow of chloride ions respectively [21]. Both mechanisms are expected to cause hyperactivity response regardless of the different target receptors. Similarly, compounds activating GABA receptors or blocking sodium channels may cause hypoactivity by enhancing the inhibitory synapses [22]. 

Based on such prior knowledge about the link between the mechanism of action and toxicological response, we defined two levels of similarity for our mixture toxicity expectation: (1) The mixture components are known to have similar target receptors or mechanism of action and (2) they show similar toxicological response (i.e., effect direction: hyper- or hypoactivity) in the STC test. Therefore, we selected mixture components based on the above factors. Compounds expected to induce hyperactivity were chlorpyrifos, chlorpyrifos-oxon and hexaconazole while abamectin, carbamazepine and propafenone are anticipated to induce hypoactivity in the STC test.

The link between effect direction and mechanism of action has been shown for single substances. In contrast, it is still open if this also works for mixture components with similar or dissimilar mechanisms of action. Therefore, the goal of the present study is to address the following questions: (1) Can the additivity models CA or IA be used to predict combined effects for neuroactive chemical mixtures when the components share a similar mode of action (hyper- or hypoativity) but show different mechanism of action? (2) Will a mixture of chemicals where the components show opposing effect direction result in an antagonistic combined effect? CA or IA cannot be used to predict the opposing effects and therefore we define antagonistic effect in this case as a counteracting effect and not a lower effect than predicted by CA or IA. We demonstrate that mixtures of neuroactive substances with different mechanisms of action follow the additivity concept and we propose ways to use the STC test in risk assessment.

## 2. Materials and Methods 

### 2.1. Test Organism

Zebrafish embryos were raised from an in-house hybrid strain (OBI-WIK strain, F3 generation). The adults were cultured under 14 h light/10 h dark photoperiod in 120 L aquaria (tap water, 26.5 ± 1 °C). Adult fish were fed twice a day either with commercial dry food flakes or *Artemia* sp. and physicochemical parameters of the aquaria water were frequently measured (pH 7–8; water hardness 2–3 mmol/L, conductivity 540–560 µS/cm, nitrate < 2.5 mg/L, nitrite < 0.025 mg/L, ammonia < 0.6 mg/L, oxygen saturation 87–91%). Spawning was initiated by inserting spawning trays 4–6 h before the end of the light cycle prior to the spawning day. Eggs were collected and cleaned 1 h after the onset of light. Fertilized embryos were selected according to Kimmel et al. [23] with a microscope and embryos between the 16th and 128th cell stage were used to start the exposure.

### 2.2. Chemicals

Chlorpyrifos (99.9%, CASRN 2921882), hexaconazole (CASRN 79983-71-4), abamectin (100%, CASRN 71751412) and propafenone-hydrochloride (CASRN 34183-22-7) were purchased from Sigma-Aldrich. Carbamazepine (99%, CASRN 298464) was purchased from Acros Organic^TM^ and chlorpyrifos-oxon (97.9%, CASRN 5598152) from Dr. Ehrenstorfer GmbH. Stock solutions were prepared in 100% dimethyl-sulfoxide (DMSO) and diluted in ISO water as specified in ISO 7346-3 (1996) (80 mM CaCl_2_·2H_2_O, 20 mM MgSO_4_·7H_2_O, 31 mM NaHCO_3_, 3.1 mM KCl). The properties, effect concentrations and model parameters for single substances used in mixture modeling are given in Table 1.

### 2.3. Mixture Testing in the STC Test

Several mixtures were designed to investigate the appropriate classification for similar and dissimilar neuroactive substances which is suitable for mixture effect prediction using CA or IA models. Mixture components were selected according to their mechanism of action and effect direction (hyper- or hypoactivity) as follows (Figure 1 and Table 2): Mixture A—compounds with the same mechanism of action and same effect direction; Mixture B—compounds with different mechanism of action but same effect direction; Mixture C—compounds in A and B; Mixture D—compounds with a different mechanism of action and different effect direction. Mixtures A and B are binary while C and D are ternary. The exposure concentrations of the mixtures given in Table 2 are based on mixture ratios of the single substances calculated as molar fraction of their effect concentrations (*EC*_50_). The *EC*_50_ concentration was selected to ensure that all components in the mixture contribute to the effect. Mixture D was particularly designed to understand if and how dissimilar compounds with different mechanisms of action and opposing effect direction would interact in the STC test. Although components of mixture D are equitoxic (in terms of *EC*_50_ ratio), the mixture was designed to reflect an unequitoxic scenario with respect to effect direction (0.33 hypoactivity: 0.66 hyperactivity).

To test if the simple case assumption of CA, i.e., substances are a dilution of each other and an equitoxic concentration of one can replace another [5], holds true for combined neurotoxicity effects in the STC test, we performed dilution experiments with the ternary mixture to simulate the hyperactivity mixtures A and B (chlorpyrifos and chlorpyrifos-oxon as well as chlorpyrifos and hexaconazole respectively). A portion of chlorpyrifos was replaced with an *EC*_50_ equitoxic portion of hexaconazole in mixture A and chlorpyrifos-oxon in mixture B (Table 2). 

The detailed procedures for STC testing have been previously reported in detail [25]. Briefly, twenty fertilized embryos were exposed in 20 mL of the mixture solution prepared from DMSO stock solution (0.1% maximum concentration) of the components, within a 60 mm glass crystallization dish covered with a watchmaker glass. Two replicates per concentration and at least 2 independent experiments were conducted. The exposed embryos were incubated at 28 °C under 14 h light/10 h dark photoperiod for 21 ± 1 h. On the next day, at 24 hpf, exposed embryos were removed from the incubator and allowed to acclimatize to room temperature for at least 30 min. Videos of normally developed embryos (without any obvious malformation) were recorded for 60 s. Collected videos were analyzed for STC counts per minute (STC frequency) by means of a workflow using the KNIME^®^ Analytical Platform [25,26].

### 2.4. Mixture Modeling

Mixture toxicity modeling was performed to investigate the capacity of concentration addition (CA) and independent action (IA) models to predict the combined effect of similar and dissimilar neuroactive substances. Effect data for the single substances used for mixture modelling were obtained from a previous study [17]. The CA mixture modeling is based on the effect concentration of the individual chemicals and it considers chemicals in a mixture to be a dilution of each other [5]. It is used to predict the mixture toxicity of chemicals with a similar mechanism of action.
(1)$$ECx_{Mix}=\overset{n}{\underset{i=1}{\sum }}\frac{P_{i}^{-1}}{ECx_{i}}\;$$

Equation (1) shows the mathematical representation of the CA model where *ECx_Mix_* is the total concentration of the mixture provoking *x* effect (i.e., 50% effect), *P_i_* is the fraction of component *i* which represents the concentration of component i in the mixture, *ECx_i_* is the concentration of component i provoking *x* effect, when applied singly. 

The IA mixture modeling is based on the effect induced by individual chemicals in a mixture. It is usually applied to predict the mixture toxicity of chemicals with the dissimilar mechanism of action.
(2)$$EC_{Mix}=1-\overset{n}{\underset{i=1}{\prod }}(1-EC_{i})$$

Equation (2) shows the mathematical representation of the IA model where *EC_Mix_* is the total effect of the mixture and *EC_i_* is the effect of component i in the mixture when applied singly. Mixture toxicity modeling was performed using an in-house excel sheet and the mixtox package in R [27].

### 2.5. Concentration–Response Modeling

Data from the mixture experiment were obtained as STC count per minute (STC frequency). The mean STC frequency was estimated for the exposed 20 embryos. The absolute STC frequency varied between the independent experiments. To combine results from independent experiments, mean percentage change in STC frequency with respect to unexposed embryos was estimated for independent experiments. Concentration–response modeling of the percentage change in STC frequency was performed using the 4-parameter logistic function (LL.4) of the drc package in R [28].
(3)$$y=c+\frac{\left(d-c\right)}{1+{\left(\frac{x}{e}\right)}^{b}}\;$$

Equation (3) shows the concentration (*x*) response (*y*) model where b is the slope; c and d are the minimum and maximum STC response set to 0 and 100, respectively; and e is the inflection point, e.g., the *EC*_50_. 

In cases of hyperactivity, the maximum effect of STC frequency was different for the three tested hyperactive chemicals—chlorpyrifos, chlorpyrifos-oxon and hexaconazole (see Figure 2). Mixture prediction using different maximal of the percentage STC effect would have been based on a non-equitoxic mixture ratio of *EC*_50_, *EC*_41_ and *EC*_24_ for hexaconazole, chlorpyrifos and chlorpyrifos-oxon respectively. To equalize the mixture ratio and maximum effect, the percentage STC change (obtained by normalizing to control) was standardized by dividing with the maximum percentage effect for each chemical to obtain a standardized percentage hyperactivity effect leading to 100% maximum effect for all hyperactive chemicals (Figure 2). This allowed us to obtain a similar half-maximum effect (*EC*_50_) for the 3 chemicals. Skipping this hyperactivity standardization step would have led to the unpredictability of mixture effects higher than that of the chemical with the least maximal effect. Scholze et al. [29] used the toxic unit extrapolation approach to equalize and extend the dose–response curves for partial agonists. However, the observed hyperactivity effect in this study is usually followed by hypoactivity (possibly due to paralysis) at higher concentrations and this could indicate a saturated hyperactive effect. This appears not to support partial agonism but rather, the differential maximal effect of the 3 chemicals could be an indication of different mechanisms of hyperactive action. A partial agonist is expected to act as an antagonist in the presence of a full agonist [30] but this was not observed in the present study. Consequently, we consider the standardized percentage hyperactivity effect to be more representative of the observations and for mixture modeling in this study. The effect concentration causing a 50% increase or decrease of the STC was estimated from the concentration–response curve and the confidence interval was estimated as 2 times the standard error.

### 2.6. Measurement of the Exposure Concentrations

Measurement of exposure concentrations was conducted to verify that test compounds were present in adequate concentrations in the test. Chemical measurement was performed only for one independent experiment of the binary mixtures since the same relation of measured and nominal concentrations were expected for other independent experiments and also for the ternary mixture. For quantifying chlorpyrifos/chlorpyrifos-oxon and chlorpyrifos/hexaconazole mixtures, chemical analyses were conducted using an HPLC system (Merck-LaChrom) with diode array (model L7450) detector. One mL of the exposure solution for each concentration of the respective mixtures was sampled and 30 µL was injected directly. A reversed-phase column (Lichrospher 60 Reverse Phase (RP) select B, Merck, C-8), with a particle size of 5 µm was used. The column temperature was set to 40 °C and the flow rate was adjusted to 0.5 mL/min. Different mobile phase ratios of AcN:water was used for chlorpyrifos/chlorpyrifos-oxon (57:43%, elution time of 15 min) and chlorpyrifos/hexaconazole (65:35%, elution time of 12 min). The substances were detected at an absorbance of 207 nm. For quantifying carbamazepine/propafenone and abamectin/propafenone mixtures, chemical analyses were performed on a linear ion trap/Orbitrap (LTQ Orbitrap XL) mass spectrometer (Thermo Scientific, Waltham, MA, USA). Samples were diluted 100 (carbamazepine/propafenone) and 10 (abamectin/propafenone) times with ISO water before injection. An Agilent 1200 series HPLC system with a Kinetex C18 column (100 × 3 mm, 2.6 µm particle size, Phenomenex) was used for chromatographic separation after injection of 10 µL of sample. We used 0.1% formic acid and methanol containing 0.1% formic acid as mobile phases at a column temperature of 40 °C and a flow rate of 0.4 mL/min. The analysis was conducted in full scan mode with a mass range of *m*/*z* 100–1000 in negative and positive mode ESI with a nominal resolving power of 100,000 (referenced to *m*/*z* 400). For peak integration, compound calibration, and compound quantification, the software program TraceFinder 3.2 (Thermo Scientific, Waltham, MA, USA) was used.

## 3. Results

### 3.1. Chemical Analysis

Results of the chemical analysis are shown in Table 3. Measured concentrations were close to the nominal concentration, typically with a maximum deviation of about 20% for the highest tested concentrations for propafenone (+37 in Hypoactive Mixture A and −3% in Hypoactive Mixture B), carbamazepine (−8.8%), chlorpyrifos (−20 and −20% in both mixtures), chlorpyrifos-oxon (+19%) and hexaconazole (+15%). Measured concentrations of abamectin were below the detection limit (MDL) in all measurements. Reasons might be due to losses or rather adsorption to the test vessels because of its high lipophilicity (logD_pH7.4(ACD/Labs)_ of 5.85). It is important to note that chlorpyrifos concentrations in DMSO stock solutions declined by 25–40% after 2 months of storage. However, this reduction in concentration did not lead to a significant difference in the STC effect (Data not shown). Therefore, we used the nominal concentrations for further mixture toxicity evaluations based on the assumption that a 20% difference between nominal and measured concentrations will not cause a significant change in the observed effect.

### 3.2. Description of Mixture Effect in Comparison to CA and IA Models

The effects of single substances used in the mixture testing have already been described in Ogungbemi et al. [17] and are summarized in Table 1. The mixture effects exceeded those of the single substances for all mixtures. Concentration–response curves for the observed and predicted mixture effects, as well as those for the single substances, are shown in Figure 3. Observed and predicted *EC*_50_ values are also shown in Table 2. 

Hyperactive Mixture A (chlorpyrifos and chlorpyrifos-oxon) (see Section 2.3 or Table 2 for the definition of the mixture name) induced hyperactivity with an *EC*_50_ of 1.25 µM. The CA and IA models were similar and they both predicted the *EC*_50_ of the mixture (Table 2). The prediction curves were within the confidence boundary of the tested mixture at low and mid concentrations but both models slightly deviated and overestimated the effect at higher concentrations (Figure 3A). The Hypoactive Mixture A (carbamazepine and propafenone) caused hypoactivity with an *EC*_50_ of 132 µM. Both CA and IA (*EC*_50_ of 159 µM and 207µM, respectively) underestimated the mixture effect. Nevertheless, CA was predictive at low and medium-high concentrations (50–150 µM) while IA was less predictive and slightly underestimated the hypoactivity effects except at the lowest concentration range up to 100 µM (Figure 3B). Overall the estimation difference was always below a factor of 2 for CA and IA.

Hyperactive Mixture B (chlorpyrifos and hexaconazole) showed hyperactivity with an *EC*_50_ of 2.79 µM (Table 2). CA could predict the exact observed *EC*_50_ of the mixture but IA slightly underestimated the mixture effect [EC_50_ = 3.69 µM] (Figure 3C). Hypoactive Mixture B (abamectin and propafenone) showed hypoactivity with an *EC*_50_ of 17.4 µM. Both CA and IA slightly underestimated the mixture toxicity with *EC*_50_ values of 23 and 27.6 µM respectively. CA aligned with the confidence boundary of the observed mixture effect while IA deviated from the observed concentration–response curve (Figure 3D). Further, we tested a ternary mixture (Mixture C comprising of chlorpyrifos, chlorpyrifos-oxon and hexaconazole). Both CA and IA models showed similar predictions and were predictive of the observed mixture effect (Figure 4). In general, we observe a trend where CA and IA could very well predict mixture hyperactivity effects but to a slightly lesser extent for the hypoactivity effects—though these differences were minor.

Further, we investigated the CA assumption that substances are dilutions of each other. Results show that substituting portions of chlorpyrifos in the Hyperactivity Mixtures A and B with hexaconazole and chlorpyrifos-oxon respectively, induced similar concentration–response curves as the non-substituted mixture (Figure 5A,B). The mixture of chlorpyrifos-oxon and (chlorpyrifos + hexaconazole) showed an *EC*_50_ of 1.77 µM which was higher than that of chlorpyrifos-oxon and chlorpyrifos mixture by only a factor of 1.4. An *EC*_50_ of 2.13 µM was estimated for hexaconazole and (chlorpyrifos + chlorpyrifos-oxon) which was lower than the hexaconazole and chlorpyrifos mix by only a factor of 1.3. 

### 3.3. Antagonistic Mixture Effects in the STC Test

Exposure of substances inducing opposing effect direction may induce antagonistic effects. Therefore, we exposed a ternary mixture of dissimilar substances (Mixture D) with different mechanisms of action and opposing effect directions (i.e., hyper- and hypoactivity). Mixtures were designed to reflect an unequitoxic scenario (0.33 hypoactivity: 0.66 hyperactivity; with respect to the corresponding *EC*_50_ values) by mixing the hypoactivity causing abamectin with two hyperactivity causing substances (chlorpyrifos and hexaconazole). The result shows that the antagonistic effect of abamectin significantly decreased the hyperactivity effect expected from hexaconazole and chlorpyrifos (Hyperactive Mixture B). Furthermore, hypoactivity effect relative to control was observed at mid-high concentration of the mixture (Figure 6). 

## 4. Discussion

In order to evaluate the mixture toxicity of neuroactive compounds, two main challenges have to be considered regarding the application of prediction models: (1) Neuroactive chemicals in mixtures interact with different biochemical targets. To capture the effects of such a mixture, a possibility is to measure the effects at converging key events. (2) Mixtures may comprise of neuroactive chemicals with opposing effects. Consequently, we explored (1) whether mixture effects of neuroactive substances with similar effect directions (whether hyper- or hypoactivity) but different mechanisms of action would be additive and if concentration addition (CA) or independent action (IA) models can predict such mixture effect and (2) if mixtures of neuroactive substances with different mechanisms/modes of action and opposing effect direction would induce observable antagonistic effects. In order to address these challenges, we used an established behavior test, the spontaneous tail coiling (STC) of zebrafish embryos. It is responsive to diverse mechanisms of actions that finally translate to increased or reduced frequency of spontaneous movements as a result of either activation or inhibition of the neuronal synapse leading to hyper- or hypoactivity respectively (STC neuroactivity hypothesis). Accordingly, we hypothesized that neuroactive chemicals inducing the same response (either hyper- or hypoactivity) in the STC test can be predicted from CA or IA models. In contrast, compounds with modes of action with opposing effects would result in antagonistic effects if compared to individual compounds. 

### 4.1. Mixture Components with Different Mechanisms of Action but Similar Effect Direction Can Act in an Additive Way

The first goal of the present study was focused on addressing the question—“Can additivity be assumed for a mixture of substances with the same mode of action (e.g., antiandrogenic) but not the same mechanism of action (e.g., receptor-blocking and inhibition of androgen production)?” which was posed in Kortenkamp et al. [31]. Based on theory, the CA model is adequate to predict mixture toxicity of similarly acting components (i.e., similar mechanisms of action) while IA is assumed to hold for dissimilarly acting chemicals. However, CA may also be applied to predict the effect of chemicals showing similar toxicological responses (i.e., hyper- or hypoactivity) or modes of action [32]. We hypothesized that irrespective of the mechanism of action, compounds inducing the same toxicological response (whether hyper- or hypoactivity) would also lead to an additive response in the STC. This allows defining the similarity/dissimilarity of mixture components based on the combined knowledge of both the mechanism of action and toxicological response. Results from the current study indicate that mixture toxicity of chemicals such as propafenone and abamectin as well as chlorpyrifos and hexaconazole that are known to induce different mechanisms of action but similar effect directions were predictable using CA and IA models. (Figure 3C,D). Predictions of the IA model were very close to those of CA and this is not surprising for a binary mixture considering that the differences between the models increase with more mixture components [33]. However, there was also no difference in the prediction of CA and IA for the ternary Mixture C (Figure 4). CA and IA models could also predict the combined effect of pyrethroids and organophosphates in a D. magna immobility assay [34]. The predictability of the mixture models for differing neuro-mechanisms as observed in zebrafish embryos and daphnids may not be applicable in other test systems or endpoints with different levels of complexity or specificity [35]. For instance, CA and IA are expected to give different predictions for simpler but specific neuro-endpoints such as neural electric signal which may not reflect an integrated output as the STC but this remains to be investigated. Therefore, it is dependent on the mechanistic understanding of the test endpoint if neuroactive substances acting on different targets in the nervous system should be considered as similarly or dissimilarly acting components [34]. This also indicates that the assessment of similarity/dissimilarity of mixture components should go beyond knowledge of molecular targets and should consider other factors such as toxicological response and secondary mode of action [36]. 

### 4.2. Mechanistic Understanding of the Predictability Power of CA and IA

The STC is presumed to be generated by depolarizations which trigger action potentials in the synapses of the primary motor neurons [37]. Consequently, it is not farfetched to consider different target interactions or mechanisms of action as similarly acting in so far as they result in the same key event (activation or inhibition of neuronal synapses) and same toxicological response (hyper- or hypoactivity). In this case, we may consider neuroactivity via the STC endpoint to be an integrated effect on neuronal synapses and CA might be more appropriate to predict mixture effects of chemicals in the STC. We showed in the present study the capacity of CA to predict mixture B (substances with different mechanisms of action but similar effect direction). This is consistent with previous studies on nervous system-related endpoints. For example, Wolansky et al. [38] found that CA was a good predictor of the mixture neurotoxicity of different pyrethroids on the motor activity of rats and Gonçalves et al. [39] reported that CA was adequate to predict the mixture effect of PAHs on fish behavior.

Based on the confidence interval of the experimental mixture, the IA model was slightly less predictive (a factor of about 1.6% deviation) for hypoactivity effects (Figure 2B,D). This could be due to unspecific effects such as axonal deformation and malformations which might contribute additional effect to the primary hypoactivity of the embryo [17]. Such additional effects would likely be captured as an integrative hypoactivity effect in the CA model. Further, the accuracy of the IA model in complex organisms such as zebrafish embryos has been questioned due to converging signaling pathways and inter-dependent subsystems [31,35,40]. For instance, Corbel et al. [10] found that carbamate and pyrethroid had a converging effect on acetylcholine concentration in the synapse even though they have different mechanisms of action. Estrogen receptor activation was also seen as an integrated effect of different cascading steroidal receptor signaling [29]. In addition, we could simulate concentration additive mixtures by replacing a portion of the mixture component with another similar acting substance (similar effect direction but different mechanism of action) (Figure 4A,B). This adds credence to the CA assumption that components can be described as a dilution of each other in the STC test. However, the results of mixture assessment with STC do not allow to favor one of the models as the differences between CA and IA were quite small. 

Mixture toxicity prediction using CA and IA models assumes that the mixture components do not interact to affect the uptake, distribution, metabolism and elimination of each other [8,41]. Mixture interaction of neuroactive substances may occur via the biotransformation pathways due to the reduced activation or competition for biotransformation sites [42]. Organophosphates were found to be a major synergistic group due to their ability to inhibit esterases which are responsible for phase 2 biotransformation of chemicals [43]. However, we did not observe synergistic interaction of a mixture of chlorpyrifos and its oxon metabolite in the present study and this could be due to potential limited biotransformation capacity of early stages of the zebrafish embryo [44] or the sensitivity of our test system. Other mixture neurotoxicity studies have shown interaction effects. For example, a mixture of chlorpyrifos and nickel on zebrafish embryos was found to be antagonistic [45] and the mixture of atrazine and chlorpyrifos was assessed as synergistic [46]. However, 120 and 96 hpf embryos, which should have higher rates for biotransformation into the active oxon metabolite, were used in these studies. 

### 4.3. Mixture Components with Different Mechanisms of Action and Opposing Effect Direction Are Antagonistic 

We investigated the STC outcome for mixtures comprising of different mechanisms of action as well as opposing effect directions (Mixture D). The results show that mixtures with both hyper- and hypoactivity-inducing components will lead to antagonistic interaction (Figure 6). Our results corroborate the recommendation of a chemical grouping for mixture analysis based on common adverse outcomes (hyper- and hypo-activity in this case) with less emphasis on the similarity of the mechanism of action [31]. Information on common adverse outcomes such as hyper- and hypoactivity will be useful to qualitatively predict mixture outcomes of multi-component/complex mixtures as well as to understand deviations from additivity. For instance, the antagonistic effects of abamectin on the hyperactivity level of the mixture of chlorpyrifos and hexaconazole (Figure 6) would have been unexplainable if only a mechanism of action-based classification was used. This particularly applies to endpoints with opposing effect directions such as locomotor activity or even gene response. For such endpoints, chemicals that primarily induce hyperactivity at low concentrations may cause hypoactivity at higher concentrations due to seizures and paralysis [13]. The use of chemicals inducing such biphasic activity as a component in a mixture without considering the primary effect direction could lead to misinterpretation of its impact on the combined effect. This biphasic activity was also observed for Mixture D in the current study and could be due to the relatively higher counteractive potency of abamectin (EC_50_ of 0.06 µM) induced at high mixture concentrations in comparison to the hyperactivity effect of chlorpyrifos and hexaconazole with much higher *EC*_50_s (Figure 6). 

Hyper- and hypoactivity response could also be used as an effect-based strategy for bio-monitoring of complex environmental mixtures which can facilitate the identification of chemicals inducing mixture neurotoxicity that would not have been detected with analytical chemical measurements [47,48]. However, equitoxic ratio of substances with opposing effect direction could lead to normalization or mitigation of the expected individual effects or mixture effects approaching control level. This counteracting effect could be a huge challenge for diagnostic risk assessment. Therefore, effect evaluation with STC as converging key event of a complex environmental mixture may only indicate an effect size related to the amount of neuroactive components if they show effect in the same direction (i.e., hyper- or hypoactivity). With opposing effects in the STC, effect evaluation may not relate to the cumulative exposure levels. However, this may present a better evaluation of the exposure level regarding the relevant biological effects and potential hazards. Nevertheless, a solution could be to spike environmental mixtures with a positive control such that deviations from the known effect size of the positive control could be an indication of the inherent effect of the mixture. In prospective mixture evaluation, one solution could be to employ a non-equitoxic mixture ratio design (e.g., 25% compound A and 75% compound B or vice versa) for opposing acting substances such that the strength of the counteracting effect is weakened. This non-equitoxic design was useful to evaluate Mixture D in the current study. However, this approach may lead to hidden effects and could give a false perspective of effect assessment. Regardless, it is necessary to elaborate on when effect normalization is an acceptable ecological risk. 

## 5. Conclusions 

We found that mixtures of neuroactive substances with different mechanisms of action but similar effect direction are additive and could be predicted using CA or IA models. Convergence and integration of effects in the nervous system provides a mechanistic understanding to support similarity classification of neuroactive compounds not only based on mechanisms of action but also considering the toxicological response or effect direction (whether hyper- or hypoactivity). Consequently, we recommend considering toxicological response or effect direction as an additional grouping factor when applying CA and IA models. On the other hand, mixtures of substances with different mechanisms of action and opposing effect direction are antagonistic. Being able to detect neurotoxicity within an environmental sample (complex mixture) is relevant since neuroactive chemicals are usually dominating concentrations of contaminants in the environment and may be major drivers of mixture toxicity. Since established effect-based tools may overlook or may not capture neurotoxicity, in this study, we propose a way to use the STC test for risk assessment despite counteracting effects which could complicate proper evaluation. 

## Figures and Tables

**Figure 1 toxics-09-00104-f001:**
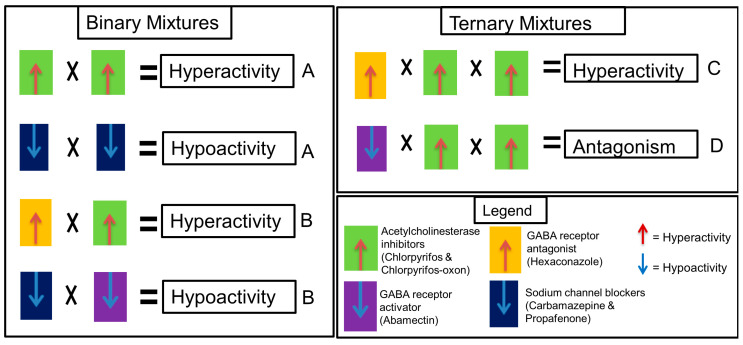
Mixture design scheme representing the hypotheses of this study. The letters A, B, C and D represent the mixture design according to Table 2. Each equation scheme for mixtures A, B and C represents a hypothesis whether concentration addition (CA) or independent action (IA) models could predict the hyper- or hypoactivity effects expected for mixtures with similar and dissimilar mechanisms of action. Equation for mixture D represents an antagonistic effect hypothesis.

**Figure 2 toxics-09-00104-f002:**
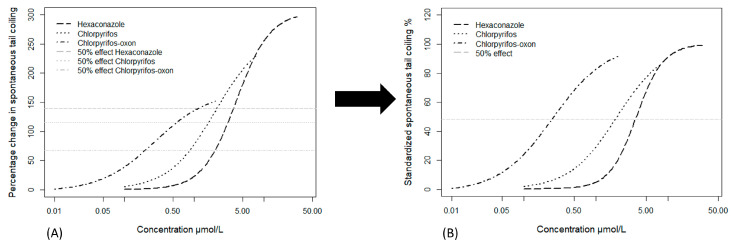
Visual representation of the data transformation for hyperactivity-inducing chemicals: (**A**) Concentration response curves showing different maximal for the hyperactivity inducing substances. The horizontal lines show *EC*_50_, *EC*_41_ and *EC*_24_ which corresponds to the 50% effect for hexaconazole, chlorpyrifos and chlorpyrifos-oxon respectively; (**B**) Standardized concentration–response curves for the hyperactivity substances. The horizontal line shows the same 50% effect for the 3 substances after standardization. Data taken from Ogungbemi et al. (2020) [17].

**Figure 3 toxics-09-00104-f003:**
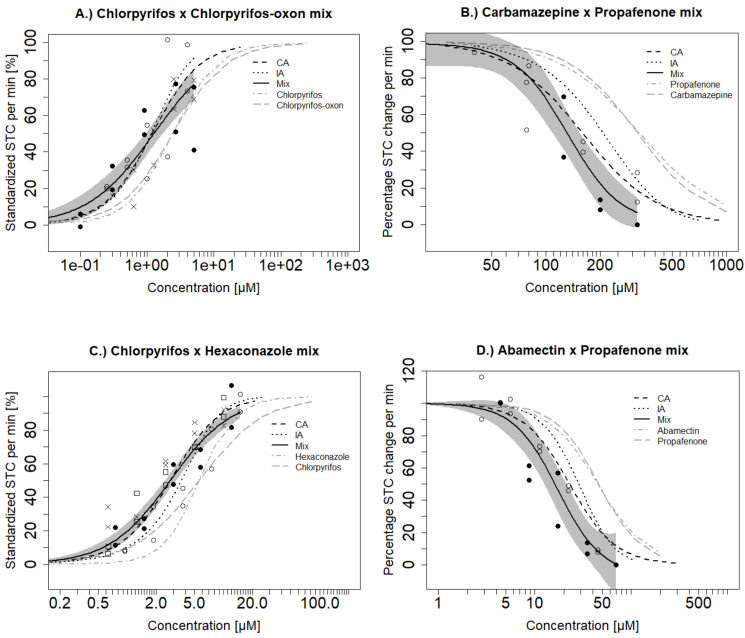
Comparison of observed (Mix) versus predicted effects of binary mixtures based on the concentration addition (CA) and independent action (IA) models in the STC. Furthermore, mixture effects are compared to single substances effects: (**A**) Hyperactivity Mixture A; (**B**) Hypoactivity Mixture A; (**C**) Hyperactivity Mixture B; (**D**) Hypoactivity Mixture B. Grey shaded areas represent the confidence interval of the fitted mixture model for the observed effect. Different symbols represent the observed mean of the STC effect for 20 embryos exposed in independent mixture experiments.

**Figure 4 toxics-09-00104-f004:**
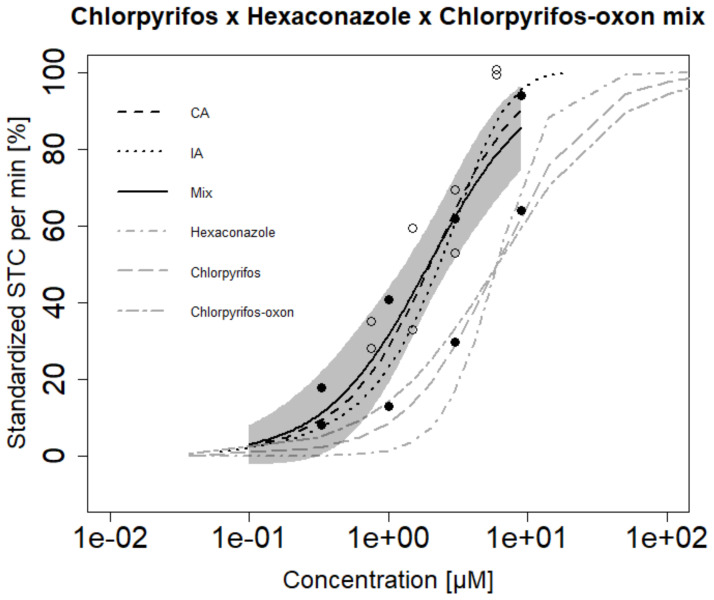
Comparison of observed (Mix) versus predicted effects of a ternary mixture based on the concentration addition (CA) and independent action (IA) models for mixture C. Furthermore, mixture effects are compared to single substances effects: Grey shaded areas represent the confidence interval of the fitted mixture model for the observed effect. Different symbols represent observed mean of STC effect for 20 embryos exposed in independent mixture experiments.

**Figure 5 toxics-09-00104-f005:**
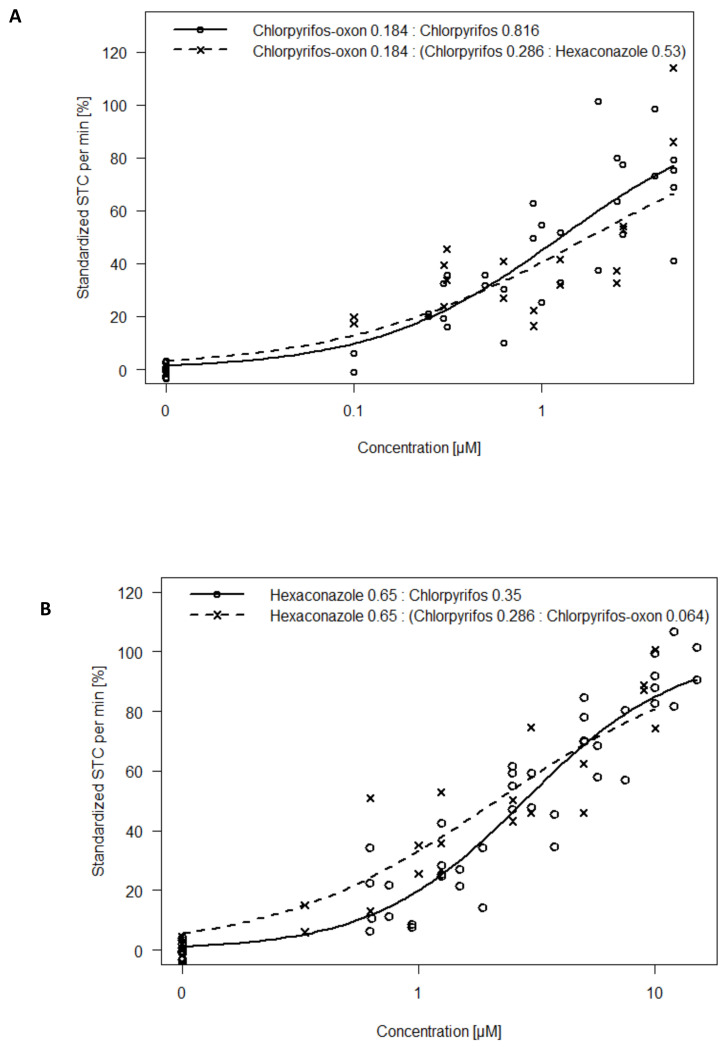
A ternary mixture is used to simulate a binary mixture by replacing a portion of one of the binary components with an equitoxic proportion of another substance: (**A**) Concentration–response curves for Hyperactive Mixture A containing chlorpyrifos-oxon and chlorpyrifos. Portions of chlorpyrifos were replaced with hexaconazole; (**B**) Concentration–response curves for Hyperactive Mixture B containing hexaconazole and chlorpyrifos. Portions of chlorpyrifos were replaced with chlorpyrifos-oxon.

**Figure 6 toxics-09-00104-f006:**
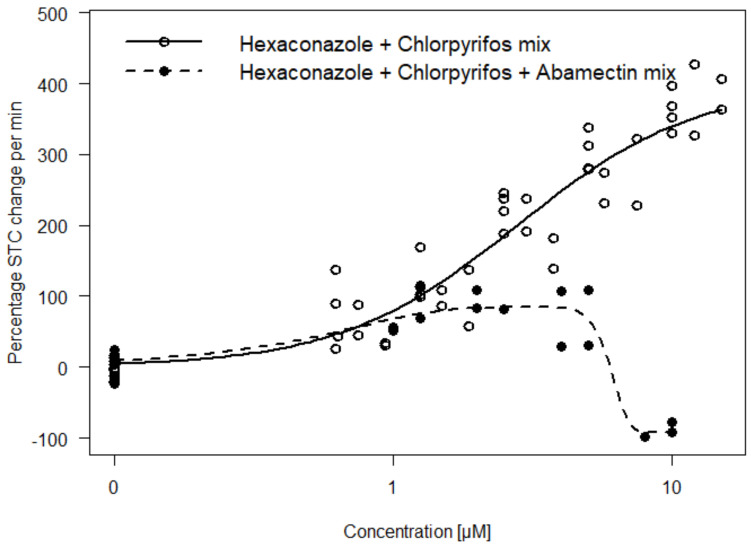
Comparison of concentration–response curves for hexaconazole and chlorpyrifos (Hyperactive Mixture B) with or without the addition of abamectin. Addition of abamectin decreases the hyperactivity effect (i.e., indicating an antagonistic effect) observed for the mixture without abamectin. A gaussian function was fitted to the data to model the biphasic effect of the mixture with abamectin.

**Table 1 toxics-09-00104-t001:** Properties and effects of single substances in the spontaneous tail coiling (STC) test.

Substance	Chemical Class	Mechanism of Action ^a^	Expected Activity, i.e., Effect Direction	STC *EC*_50_ (µmol/L) ^b^	Slope of crc ^b^
Chlorpyrifos	Organophosphate	Acetylcholinesterase inhibitor *	Hyperactivity	1.85 (1.95)	1.30
Chlorpyrifos-oxon	Organophosphate	Acetylcholinesterase inhibitor *	Hyperactivity	0.32 (0.44)	1
Hexaconazole	Triconazole	Ergosterol biosynthesis inhibitor *	Hyperactivity	4.03 (3.63)	1.80
Abamectin	Avermectin	Activation of GABA-gated chloride channel ^$^	Hypoactivity	0.06 (0.09)	1.70
Carbamazepine	Dibenzazepine	Sodium channel blocker ^#^	Hypoactivity	271	2.28
Propafenone	Aromatic Ketone	Sodium channel blocker ^#^	Hypoactivity	32 (46)	1.94

^a^ Mechanism of action was obtained from different sources including ^#^
http://drugbank.com * pesticide properties database (https://sitem.herts.ac.uk/aeru/ppdb/index.htm) and ^$^ Sánchez-Bayo, (2012) [24]; ^b^ Data obtained from Ogungbemi et al., (2020), the minimum and maximum of the concentration–response curves (crc) were set to 0 and 100, respectively. Values in parenthesis were obtained from independent experiments and were used for the mixture modelling.

**Table 2 toxics-09-00104-t002:** Summary of the mixture design, observed toxicity and predicted toxicity.

Mixture	Substances	Observed Activity	Mixture Ratio ^a^	Exposure Concentration (µmol/L) ^b^	Predicted EC_50_ (µmol/L)		Observed *EC*_50_ (µmol/L)
					CA	IA	
**Mixture A**	Chlorpyrifos and chlorpyrifos-oxon	Hyperactivity	0.816:0.184	0, 0.25, 0.5, 1, 2, 4 0, 0.1, 0.3, 0.9, 2.7, 5 0, 0.313, 0.625, 1.25, 2.5, 5	1.19	1.16	1.25
	Carbamazepine and propafenone	Hypoactivity	0.86:0.14	0, 40, 80, 160, 320 0, 78, 125, 200, 320	159	207	132
**Mixture B**	Hexaconazole and chlorpyrifos	Hyperactivity	0.65:0.35	0, 0.94, 1.87, 3.75, 7.5, 15 0, 0.75, 1.5, 3, 5.73, 12 0, 0.625, 1.25, 2.5, 5, 10 0, 0.625, 1.25, 2.5, 5, 10	2.79	3.69	2.79
	Abamectin and propafenone	Hypoactivity	0.002:0.998	0, 2.8, 5.6, 11.3, 22.5, 45 0, 4.38, 8.75, 17.5, 35, 70	23	27.6	17.4
**Mixture C**	Chlorpyrifos, hexaconazole and chlorpyrifos-oxon	Hyperactivity	0.603:0.324 :0.073	0, 0.75, 1.5, 3, 6, 12 0, 0.33, 1, 3, 9	2	2.19	1.95
**Mixture D**	Chlorpyrifos, hexaconazole and abamectin	Hyper and Hypoactivity	0.34:0.64 :0.02	0, 1.25, 2.5, 5 0, 1, 2, 4	- *	-	-
**Simulation of Hyperactive Mixture A**	Chlorpyrifos-oxon, (chlorpyrifos and hexaconazole)	Hyperactivity	0.184:(0.286 :0.53)	0, 0.313, 0.625, 1.25, 2.5, 5 0, 0.1, 0.3, 0.9, 2.7	-	-	-
**Simulation of Hyperactive Mixture B**	Hexaconazole, (chlorpyrifos and chlorpyrifos-oxon)	Hperactivity	0.65:(0.286 :0.064)	0, 0.625, 1.25, 2.5, 5, 10 0, 0.33, 1, 3, 9	-	-	-

* no mixture and toxicity predictions; ^a^ Mixture ratios are calculated as molar fraction of the total concentration. The ratio in the mixture is defined by the ratio of *EC*_50_s. ^b^ The given exposure concentrations refer to the exposure range of independent experiments. In subsequent experiments, often different ranges were used to promote a better description of concentration–response curves. All concentration ranges were combined for concentration–response modelling.

**Table 3 toxics-09-00104-t003:** Measured concentrations of single substances in each mixture in micromole/liter. Values in round brackets are the percentage change of the measured concentrations with respect to the nominal concentrations while values in squared brackets are nominal concentrations that are below detection limit.

Hyperactive Mixture A		Hypoactive Mixture A		Hyperactive Mixture B		Hypoactive Mixture B	
Chlorpyrifos	Chlorpyrifos-Oxon	Carbamazepine	Propafenone	Chlorpyrifos	Hexaconazole	Abamectin	Propafenone
<MDL [0.25]	<MDL [0.05]	92.2 (+36)	22.1 (+120)	<MDL [0.2]	0.4 (−4)	<MDL [0.009]	6.0 (+37)
0.2 (−59)	<MDL [0.1]	128.0 (+20)	33.1 (+89)	0.2 (−50)	0.8 (+5)	<MDL [0.018]	11.4 (+31)
0.7 (−32)	0.5 (+109)	190.8 (+11)	47.7 (+70)	0.6 (−37)	1.8 (+10)	<MDL [0.035]	20.2 (+15)
1.8 (−12)	0.6 (+39)	250.7 (−8.8)	61.3 (+37)	1.4 (−23)	3.6 (+10)	<MDL [0.07]	31.4 (−10)
3.2 (−20)	1.1 (+19)			2.8 (−20)	7.5 (+15)	<MDL [0.14]	68.0 (−3)

MDL = Method detection limit. Chlorpyrifos MDL = 0.1 µM, Chlorpyrifos-oxon MDL = 0.1 µM, Hexaconazole MDL = 0.3 µM, Carbamazepine MDL = 0.0045 µM, Propafenone MDL = 0.0034 µM, Abamectin MDL = 0.0005 µM.

## Data Availability

The data presented in this study are openly available in Zenodo at https://doi.org/10.5281/zenodo.4640059.

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
