# Peer review of "Assessing Combined Effects for Mixtures of Similar and Dissimilar Acting Neuroactive Substances on Zebrafish Embryo Movement"

_toxics, 2021, doi:10.3390/toxics9050104_

Round 1

Reviewer 1 Report

General comments:

In the manuscript entitled “Assessing combined effects for mixtures of similar and dissimilar acting neuroactive substances on zebrafish embryo movement”, the authors evaluated the cocktail effect of four types of mixtures, in total four binaries and two ternaries, including six neuroactive substances   with considering their similar and dissimilarly mechanisms of action and modes of action. Ecotoxicology studies on the applicability of the conventional CA and IA models to predict the combined effects have been frequently focused on lethal endpoints rather than non-lethal ones such as spontaneous tail coiling in the light of the neurotoxicity. Thus, the manuscript presented by the author can give good knowledge to the audience in the related fields.

In the manuscript, the authors concluded that the mixtures of neuroactive substances with different mechanisms of action but similar effect direction (mode of action) are additive and could be estimated by the CA or IA. In addition, mixtures of substances with different mechanisms of action and opposing effect direction are antagonistic. This findings seem to follow the funnel hypothesis: many interactions occur in a complex mixture, overall combined toxicity shifts in both directions and thus ultimately cancelling each other out (Warne & Hawker 1995; Kortenkamp A, Backhaus T, Faust M. 2009).

  • Warne, M. St. J. & Hawker, D. W. 1995, "The number of components in a mixture determines whether synergistic and antagonistic or additive toxicity predominate: The funnel hypothesis", Toxicology and Environmental Safety, vol. 31, pp. 23-28.

However, the authors tested their hypothesis with small number of neuro active compounds and their mixtures basically while there are a plethora of neuroactive substances. In this context, the mixture toxicity design used in this manuscript needs to be more clearly described. I recommend the major revision for the manuscript for the publication in Toxics.

Specific comments:

Section 2.3 on page 3: More clearly explain why there is no target binary mixture having different mechanisms of action, but opposing effect direction in Table 2.

Section 2.5 on page 5: Describe more clearly how the non-equixotic mixture ratio (e.g, EC50, EC41, and EC24) were determined to equalize the mixture ratio and maximum effect.

Table 2 on page 4: In Table 2, describe more clearly how the mixture ratios were determined. What do you mean by “Highest exposure”?

Section 3 on page 7: what do you mean by “propafenone (3 and 37%)” and “chloropyrifos (20 and 20 %)”?

Section 4.3 on page 13: Figure 5B cannot be found in the manuscript.

Reviewer 2 Report

This experiment explored whether the actions of toxins with different neural targets would result in response modes in the spontaneous tail coiling response of embryonic zebrafish, contributed by additive effects (concentration addition), or by more complex and potentially contradictory effects (independent action). It attempts to validate this animal model for toxin mixtures. Embryos were exposed at the 16-128 cell stage, but only “normally developing” (which was not defined) embryos’ behavior was assessed at 24 hrs hfp. Exposures to 6 toxins followed 4 regimes of mechanism of action, and the concentration of each was determined by the effect conc. of each toxin alone.

The work is timely and worthwhile in its uncommon goal to demonstrate behavioral frequency effects of progressively higher concentrations of mixtures of toxins with the intent to observe whether these effects match models of the toxins’ physiological effects (neuroactive vs neuroinhibitory).  If the observed dose-effect curve fits closer to the predicted/modeled CA or IA curve, the toxin mixture can be assumed to have effects similar to the model it matches most closely. But important details are missing that enable a reader to understand exposures and results. Tables and figures and prose lack a common language for referring to elements of the experiment. Methods necessary to understanding the experiment are missing. Tables are missing labels that are required to understand what was tested. Figures containing results are not referred to where the prose results occur, and their captions fail to make up for inadequacies in reporting the methodology. The discussion is diffuse and general, and does not sufficiently address the specific results presented. These omission sum to make reader reception of the results uncertain. Specifics of the type of missing details are described below.

Para under title 2.5 concentration response modeling, p. 5:

Did exposed and unexposed embryos that were used to calculate the percent change in a toxin originate from the same batch of embryos? This would be important for accepting this ratio, given that earlier the authors report that absolute STC frequency varied between experiments. The 4th paragraph on p. 5 uses the phrase “unexposed embryos from independent experiments” which encourages doubt that the exposed and unexposed embryos were from the same batches. Were they the same, or different batches? If different, please reassure a reader that absolute frequencies in different experiments were at some acceptable range that is stated explicitly. E.g. if we are comparing x to 1.1x, there is little concern.

I believe having a hyperactive and hypoactive Mixture A, and a hyperactive and hypoactive Mixture B is confusing, especially since there is no relationship between most of the components in the A’s and in the B’s. Toxins can all be considered unique, and they do not share e.g. EC50s. What is the purpose of hyper and hypo A’s and B’s, why not give them all unique labels?

Table 2. The caption is inadequate. This table’s last column contains observed EC50.

I am not sure I understand the concentration of mixtures studied. The first mixture A contains a ratio of 0.816:0.184, but by what - weight, molarity? - of chlorpyrifos:chlorpyrifos oxon, and is predicted to have an EC50 of 1.16-1.19 µM depending on the model. Its highest concentration used was 5. How does one arrive at 5 µM or any tested mixture concentration from the components? These seem to be from approximately 0.01 µM to 10 µM (looking at the first symbols in Fig. 2A), but are different for other mixtures. Why are concentration ranges not stated in methods?

Table 3

To what does each row of the table refer (lacks label)? There is a steady increase in measured concentration at each successive row, and 2nd paragraph under section 3.2 suggests these are different concentrations of mixtures tested (“low”, “mid”), but methods do not describe this.

Fig 2 seems to refer to results of the uniform hyper- or hypoactive mixtures A and B, although the caption does not state this, and illustrates that observed mixture behavior is left-shifted compared to single compound behavior, as expected of additive effects. The line for “mix” is the observed behavior frequency but this is not stated in the caption, moreover, in Table 2, the mixture’s EC50 is referred to as “observed”, not “mix” (an example of not-in-common languages between authors of the tables vs figures). Please be consistent, and state the chosen convention for referring to elements of the experiment in the caption, and in the methods and results prose. Fig. 2 is not cited at the same time as results are reported in Results prose, but rather later in the paragraph that contains the observed EC50s. Is each symbol/concentration the mean of behavior in 20 embryos? Please be clearer.

The authors should explain why CA and IA models often fall so closely together on the dose-effect curve without actually being the same, for example, predicted CA and IA are close together in Fig. 2A and 2C, and in Fig. 3, essentially because the illustrated toxins are uniformly hyper- or hypoactive mixtures. Other uniformly acting toxins’ modeled CA and IA are further apart. Why?

Fig. 5.

IA toxins may work antagonistically, with hyper- and hypoactivity canceling out one another, especially if they are carefully calibrated mixtures as they are here. These should produce a distinctly different curve from CA toxins. I do not agree with the authors’ statement (p. 13) that: “ … mixtures with both hyper- and hypoactivity inducing components will lead to antagonistic interaction” In Fig. 5, the rightmost 2 solid points are clearly driving the fit of the curve; without them the +abamectin curve would more closely resemble the mixture without (-)abamectin, and in any case, as illustrated, +abamectin is not markedly right-shifted as one would expect of a distinct form of action relative to -albamectin. Why does the +abamectin mixture so closely track the -abamectin up to a little more than 1 µM, when the EC50 for abamectin is 0.06 µM (from Table 1)? The authors themselves pose (p. 3): Will a mixture of chemicals where the components show opposing effect direction result in an antagonistic combined effect? The answer seems to be a qualified no, but an explanatory hypothesis is required.

Minor

Context for what is physiologically or environmentally relevant about these toxins should be included in the introduction.

Round 2

Reviewer 1 Report

I would suggest that the manuscript can be accepted in Toxics.

Reviewer 2 Report

The manuscript has been improved in editing by the authors.